# A Survey on the Use of Spirometry in Small Animal Anaesthesia and Critical Care

**DOI:** 10.3390/ani12030239

**Published:** 2022-01-19

**Authors:** Mathieu Raillard, Olivier Levionnois, Martina Mosing

**Affiliations:** 1AniCura Regiondjursjukhuset Bagarmossen, Ljusnevägen 17, 128 48 Bagarmossen, Sweden; 2School of Veterinary Science, Faculty of Science, The University of Sydney, Evelyn Williams Building No. B10, Sydney, NSW 2006, Australia; 3Anaesthesiology Section, Department of Clinical Veterinary Sciences, Vetsuisse Faculty, University of Berne, 3012 Berne, Switzerland; olivier.levionnois@vetsuisse.unibe.ch; 4School of Veterinary and Life Sciences, College of Veterinary Medicine, Murdoch University, 90 South Street, Murdoch, WA 6150, Australia; M.Mosing@murdoch.edu.au

**Keywords:** compliance, resistance, spirometry, survey, tidal volume, ventilation

## Abstract

**Simple Summary:**

Spirometry can be used to monitor airway pressures, flows, and volumes. Its relevance in small animal anaesthesia is documented. The way the Veterinary Anaesthesia and Intensive Care communities use spirometry was not found in the literature. The objective was to document the use of spirometry and ventilation settings in small animal anaesthesia and intensive care through a descriptive, open, online, anonymous survey. The survey was advertised on social media and via email. Participation was voluntary. The google forms platform was used. It consisted of eight sections in English. Simple, descriptive analyses were applied. There were 128 respondents. Respondents used spirometry more when dogs were mechanically ventilated as opposed to spontaneously breathing. Over 3/4 of the respondents considered spirometry essential in “selected” (43%) or “most” cases (33%). Multiple devices and technologies were used. The majority of the respondents were not directly involved in or informed about the calibration of their equipment. More information on variables monitored and technical background on spirometers is required.

**Abstract:**

The objective was to document the use of spirometry and ventilation settings in small animal anaesthesia and intensive care through a descriptive, open, online, anonymous survey. The survey was advertised on social media and via email. Participation was voluntary. The google forms platform was used. It consisted of eight sections in English: demographic information, use of spirometry in spontaneously ventilating/mechanically ventilated dogs, need for spirometry, equipment available and calibration status, ventilation modes, spirometry displays, compliance (C_RS_) and resistance (R_RS_) of the respiratory system. Simple descriptive analyses were applied. There were 128 respondents. Respondents used spirometry more in ventilated dogs than during spontaneous breathing. Over 3/4 of the respondents considered spirometry essential in “selected” (43%) or “most” cases (33%). Multiple devices and technologies were used. The majority of the respondents were not directly involved in or informed about the calibration of their equipment. Of all displays, pressure-volume loops were the most common. Values of C_RS_ and R_RS_ were specifically monitored in more than 50% of cases by 44% of the respondents only. A variety of ventilation modes was used. Intensivists tend to use smaller VT than anaesthetists. More information on reference intervals of C_RS_ and R_RS_ and technical background on spirometers is required

## 1. Introduction

Spirometry can be used to monitor continuously airway pressures, flows, and volumes and calculate compliance (C_RS_) and resistance (R_RS_) of the respiratory system. Its relevance in small animal anaesthesia is documented [1]. The way the Veterinary Anaesthesia and Intensive Care communities use spirometry was not found in the literature. We investigated this through an online survey.

We had several goals. Firstly, we wanted to gather some habits of the community on the use of and perceived need for spirometry. Secondly, we wanted to obtain an overview of the equipment available and its calibration status. Thirdly, we wanted to investigate what spirometry displays were frequently monitored. Fourthly, we wanted to find out whether participants specifically considered C_RS_ and R_RS_. Given the effects of ventilation modes on C_RS_ [2] and R_RS_ in anaesthetized dogs, information on commonly used ventilation settings was also sought. The parts of the survey that were not equipment-related (by nature, non-species specific) focused on dogs.

## 2. Materials and Methods

### 2.1. Survey Design

The CHERRIES (Checklist for Reporting Results of Internet E-Surveys) statement was followed [3]. This was a descriptive, open, anonymous survey. Convenience sampling (a method of selection from a conveniently available pool of respondents) was used.

### 2.2. Institutional Review Board Approval

Ethics were managed under the requirements of the University of Bern, where the research was undertaken. The study was conducted according to the guidelines of the Declaration of Helsinki. Ethical approval was waived for this study after review from the local ethical committee (Gesundheits-, Sozial- un Integrationsdirektion Kantonale Ethikkommission für die Forschung, Kanton Bern) because this project was not subject to any authorization according to national law (Humanforschungsgesetz, Art. 2, Abs 1).

### 2.3. Informed Consent

The introduction page informed the participants on the objectives, number of questions, and expected time needed to answer the survey. It also reported: “The data collected will remain confidential. No information to identify you or your practice will be collected. (...) You are undertaking this survey on a voluntary basis. You can decide at any time that you no longer want to participate. (...) By clicking “NEXT” you certify that you have read and understood the purpose of this study, your rights as an informed participant and that your answers will be used for analysis and publication.”

### 2.4. Development and Pretesting

The survey was developed to be comprehensive and easily filled. It was sent to two ECVAA diplomates (European College of Veterinary Anaesthesia and Analgesia) with interest in respiratory mechanics, and one biomedical engineer with expertise in veterinary anaesthesia. The Survey was altered following their feedback (unclear content was reformulated, some questions were added/removed).

### 2.5. Recruitment Process and Description of the Target Population

The survey was advertised on social media (Facebook, LinkedIn, and Twitter) on the pages of the AVA (Association of Veterinary Anaesthetists), SEAAV (Sociedad Española de Anestesia y Analgesia Veterinaria), Anaesthesia and Emergency and Critical care chapters of the ANZCVS (Australia and New Zealand College of Veterinary Scientists), and EVECCS (European Veterinary Emergency and Critical Care Society). Links were sent using a shared email list of the international Veterinary Anaesthesia and Analgesia community (“acva-list”), the EVECCS newsletter, and through the secretaries of the ANZCVS Anaesthesia and Emergency and Critical care chapters and the ECVAA. Other institutions (including the American College of Emergency and Critical Care, ACVECC) were also contacted. A selection of survey announcements is presented in Appendix A.

### 2.6. Survey Administration

Google forms was used to host the survey. Participation was voluntary; no incentives were offered. The survey was conducted between February and July 2018. There were eight sections, composed of two to eight items each (39 multiple choice or open-ended questions, all in English). Items were not randomized. Adaptive questioning was not used, although some questions were facultative. Respondents could review and change their answers through the “Back” button until they submitted the full questionnaire.

No effort was made to prevent multiple entries from the same individual (no IP check or specific cookies used). Responses were received only if the respondents completed the final screen of the survey and submitted their questionnaire. View rate, participation, and completion rates were not calculated.

### 2.7. Questionnaire

Section one asked demographic information (three questions), including background, working environment, and country. Sections two and three focused on the frequency of the use of spirometry (intended as a measurement of airway flows and/or inference of respiratory volumes) in spontaneously ventilating and mechanically ventilated dogs. Section four asked if respondents considered spirometry “essential”. Section five asked what spirometry devices were used and about equipment maintenance/calibration. Section six focused on ventilation modes used and initial setup, including inspiratory pressure or tidal volume (VT) and adaptation to body condition score. Section seven gathered information on spirometry displays used, and, section eight, on C_RS_ and R_RS_.

The questionnaire is available on request to the corresponding author.

### 2.8. Statistical Analysis

Time to answer the survey was not monitored. Every questionnaire was analysed. Completeness was checked for each question. Items were not weighted. Simple descriptive analysis was used; the percentage rate was calculated.

For VT analysis, respondents were categorized as either “anaesthetists” or “intensivists” according to their answer on which professional background (diploma, practice) they declared. Responses from people working in both specialties, blank responses, and “not applicable” were not included. Distribution of proposed VT values was checked using the d’Agostino normality test. Medians of VT were visually compared. To check whether intensivists and anaesthetists tended to use different VT, the Mann–Whitney *U*-test was performed (the middle of the range was arbitrarily taken as a single VT value for responses reporting a range rather than a single value). Varga and Delanay’s A was used to investigate the effect size. Successively, VT was defined as “low”, 8 mL kg^−1^ or below, and “high”, 12 mL kg^−1^ or above; if a range was centred around 8 or 12 mL kg^−1^, the lowest value was considered for the lower end of VT and the highest for the higher end so that “6 to 10 mL kg^−1^” would be classified as “low” and “10 to 15 mL kg^−1^” would be classified as “high”. The chi-square test (with Yates’ continuity correction) was used to test for differences in low versus high VT use between intensivists and anaesthetists. Analyses were performed using R V. 1.3.5 Mac OS (The R Foundation for Statistical Computing, http://www.R-project.org, Austria, accessed on 17 January 2022). A *p* value < 0.05 was considered significant.

## 3. Results

Section one, demographic information: There were 128 respondents (Table 1).

Sections two, three, and four, use of spirometry: In spontaneously ventilating dogs, spirometry was “never used” by 38/127 (30%) respondents [25/38 had no spirometer available; 13/38 had spirometers but were not using them] (no answer: 1/128). Spirometry was used by 27/127 (21%) of the respondents “in less than 25% of cases”, 11/127 (9%) “in 25 to 50% of cases”, 19/127 (15%) “in 50 to 75% of cases”, and 32/127 (25%) “in 75 to 100% of cases”.

In mechanically ventilated dogs, spirometry was “never used” by 24/127 (19%) respondents [18/24 had no spirometer available, 6/24 had access to spirometers] (no answer: 1/128). Spirometry was used by 15/127 (12%) of the respondents “in less than 25% of cases”, 10/127 (8%) “in 25 to 50% of cases”, 17/127 (13%) “in 50 to 75% of cases”, and 61/127 (48%) “in 75 to 100% of cases”.

All the 24/127 respondents that were not using spirometry in ventilated dogs also answered that they did not use spirometry in spontaneously ventilating dogs.

Spirometry was considered essential “in no case” by 11/128 respondents (9%), “in specific cases” by 55/128 respondents (43%), “in many cases” by 19/128 respondents (15%), and “in most cases” by 43/128 respondents (33%).

Section five, equipment and calibration: Most respondents (64/120, 53%) used the D-Lite/Pedi-Lite flow sensor (Pitot tube, Datex-Ohmeda/GE Healthcare). Wright respirometer, Spiromed (Dräger), or NICO (Respironics) were used by 18, 5, and 3 respondents, respectively. Other technologies, including heated wire anemometer, variable orifice flow sensor, ultrasonic flow sensor, and fixed orifice flow sensor (mostly non-specified), were used by 9, 7, 4, and 3 respondents, respectively. The ventilator display was used by 51/120 respondents (43%) to monitor spirometry. A variety of ventilators were used to monitor spirometry, but many respondents did not provide detail on their device(s) or on the technology behind the measurements. Four respondents used at least one additional technique to the above monitors/technologies but did not specify which. Six respondents did not know which technology they were using, of whom five considered spirometry necessary “in many” or “in most cases”. No answer: 8/128.

When asking about the verification of the calibration of spirometry devices, 65/119 (55%) declared that it was performed by the manufacturer when the device was serviced; 15/119 (13%) that it was regularly done by an anaesthesia team member; 14/119 declared that it was never checked (12%); and 25 (21%) reported they did not know. No answer: 9/128.

Section six, Ventilation settings: Pressure modes were used by 94/124 respondents (76%), volume modes by 104/124 (84%), and dual modes (e.g., pressure-regulated volume-controlled ventilation) by 49/124 (40%) (Multiple answers were possible; no answer: 4/128).

Regarding pressure modes, 10/128 respondents did not answer or did not use them. The initial pressure was set ≤ 10 cmH_2_O by 25/118 (21%) respondents (including 8 cmH_2_O by 11/118), at 10 cmH_2_O by 41/118 (35%), to a range centred around 10 cmH_2_O (pressure ranges: 9–11, 8–12, 5–15, 0–20) by 5/118 (4%), to a range ≥ 10 cmH_2_O by 47/118 (40%) (including 16/118 setting up 12 cmH_2_O and 6/118, 15 cmH_2_O). The pressure setting was adapted to size, breed, morphology, and BCS by 73/123 (59%) respondents (No answer: 5/128).

Regarding volume modes, 9/128 did not answer or did not use them. The initial VT was set ≤ 10 mL kg^−1^ by 21/119 (18%) respondents, at 10 mL kg^−1^ by 66/119 (55%), between 10 to 20 mL kg^−1^ by 31/119 (26%) (including 10 using 12 mL kg^−1^ and 5 using 15 mL kg^−1^), and between 8 to 12 mL kg^−1^ by 1/119. The VT was adapted to size, breed, morphology, and body condition score (BCS) by 74/123 (58%) respondents (No answer: 5/128).

Median VT initially set by intensivist and anaesthetists was 9 mL kg^−1^ and 10 mL kg^−1^, respectively (*p* < 0.01; effect size was 0.78 [large]). A higher proportion of intensivists used “low” VT and a higher proportion of anaesthetists used “high” VT, λ^2^(1, *N* = 40) = 15.09, *p* < 0.01.

Section seven, spirometry displays: Pressure-volume loops were used by 89/117 respondents (76%), flow-volume loops by 51/117 (44%), pressure-time curves by 46/117 (39%), flow-time curves by 38/117 (32%), and volume-time curves by 20/117 (17%) (Multiple answers were possible; no answer: 11/128). A total of 18/117 respondents declared using no curve but only numbers (15%).

Section eight, C_RS_ and R_RS_: Spirometry was “never used” specifically to monitor C_RS_ and R_RS_ by 26/125 (21%) respondents, used in “less than 25% of cases” by 34/125 (27%), used “in 25% to 50% of cases” by 10/125 (8%), used “in 50% to 75% of cases” by 17/125 (14%), and used “in 75% to 100% of cases” by 38/125 (30%) (no answer: 3/128).

Expected physiologic numerical values of C_RS_ and R_RS_ were extremely variable [80/128 (63%) and 106/128 (83%) respondents did not answer or stated they did not know].

A selection of results is summarized in Figure 1.

## 4. Discussion

Results of this survey illustrate that a variety of spirometry technologies was used in dogs, more frequently during mechanical ventilation than in spontaneous breathing. Most respondents were not involved in or informed about the calibration of their equipment. Initial settings of mechanical ventilation varied widely. The shape of pressure-volume loops was the most commonly assessed display. Values of C_RS_ and R_RS_ remain unknown.

Over 3/4 of the respondents considered spirometry essential in “selected” or “most” cases. However, one-third of them was unaware of the calibration status of their equipment and over half relied exclusively on the service. The American Thoracic Society and the European Respiratory Society recommend daily checks of diagnostic spirometers (single discharge of a 3-L calibration syringe) and recalibration if the reading was out of the ±3% range [4]. The same accuracy cannot be expected from spirometers used in anaesthesia (continuous monitoring, variation in gas composition). The manual of use for GE modules reports that the need for calibration is “infrequent” and that calibration should be performed only if the accuracy of volumes obtained was out of ±6%. No evidence was found on the actual performance over time of spirometers used in anaesthesia. This should be investigated.

Respondents used the shape of the Pressure-Volume loop more than C_RS_ and R_RS_ absolute values. Pressure-volume loops were first described as sigmoid static curves [5]. Dynamic proximal (to the endotracheal tube) loops might be more useful than static curves to assess compromised airway patency, incorrect placements of the orotracheal tubes, leaks, or changes in C_RS_ and/or R_RS_ [6]. The resistance generated by the endotracheal tube can make their interpretation difficult [7]. Dynamic Pressure-Volume loops allow the monitoring of trends rather than precise measurements. The shape of such curves widely depends on the scale of the graph displayed, which can be altered or automatically adjusted by the monitors. Therefore, clinical interpretation of the respiratory mechanics could be incorrect. Values of C_RS_ and R_RS_ are independent of the scale of the curves displayed. In this survey, Pressure-Volume loops were used by one-third of the respondents in >75% of the cases. However, nearly 50% of all respondents did not use numerical values at all or used them in less than 25% of the cases. To date, equations only accounting for body weight have been reported for C_RS_ and R_RS_ values in healthy dogs [8,9]. Additional information and reference intervals are necessary to allow objective data interpretation. Additionally, C_RS_ and R_RS_ values can vary depending on the ventilation mode used [2].

Although nearly 60% of people adjusted their VT or inspiratory pressures in function of size, breed, morphology, and BCS, over half of the respondents set the initial VT at 10 mL kg^−1^ in volume-controlled ventilation. While (unreferenced) recommendation exists to start volume-controlled mechanical ventilation with VT below 10 mL kg^−1^ [10], evidence suggests that higher VT might be more appropriate in healthy dogs [11]. Interestingly, intensivists were more likely to use VT below 8 mL kg^−1^ and anaesthetists to use VT above 12 mL kg^−1^. This may originate from differences in the animals’ presentation in the different specialties (i.e., mostly healthy lungs in anesthesia vs. pathologic changes in ECC). It may also be that anesthetists and intensivists base their clinical decisions on different scientific literature [11,12]. Therefore, it might be interesting to differentiate the specialties in studies investigating respiratory mechanics.

This study has several limitations. Firstly, the number of respondents was low, and intensivists were under-represented. This could be due to the fact that the right channels were not used to contact the ECC community. This certainly limits the interpretation of our findings. Secondly, convenience sampling was used. Selection bias is likely: Respondents were probably interested in the topic to answer the survey. Thirdly, multiple responses from the same individual could not be excluded. Fourthly, some respondents stated they never used spirometry but gave detail on monitors and curves used. Possible explanations include either an error in entering the information or that these individuals work at centres where spirometers are available, but they do not use them clinically and interpreted the question asked in the survey in this regard. Fifthly, there were differences in the number of respondents not having access to a spirometer between spontaneous and ventilated dogs. This may reflect different contexts of spontaneous vs. mechanical ventilation and equipment availability.

## 5. Conclusions

Spirometry was used more frequently in mechanically ventilated dogs than during spontaneous ventilation by the respondents of this survey. The shape of pressure-volume loops was the most common display. A variety of technologies and ventilation modes were used. Future investigations should (1) assess the accuracy of spirometers commonly used in veterinary anaesthesia and (2) determine the reference intervals for C_RS_ and R_RS_ in healthy, anaesthetized dogs, as this survey confirmed the need for such information.

## Figures and Tables

**Figure 1 animals-12-00239-f001:**
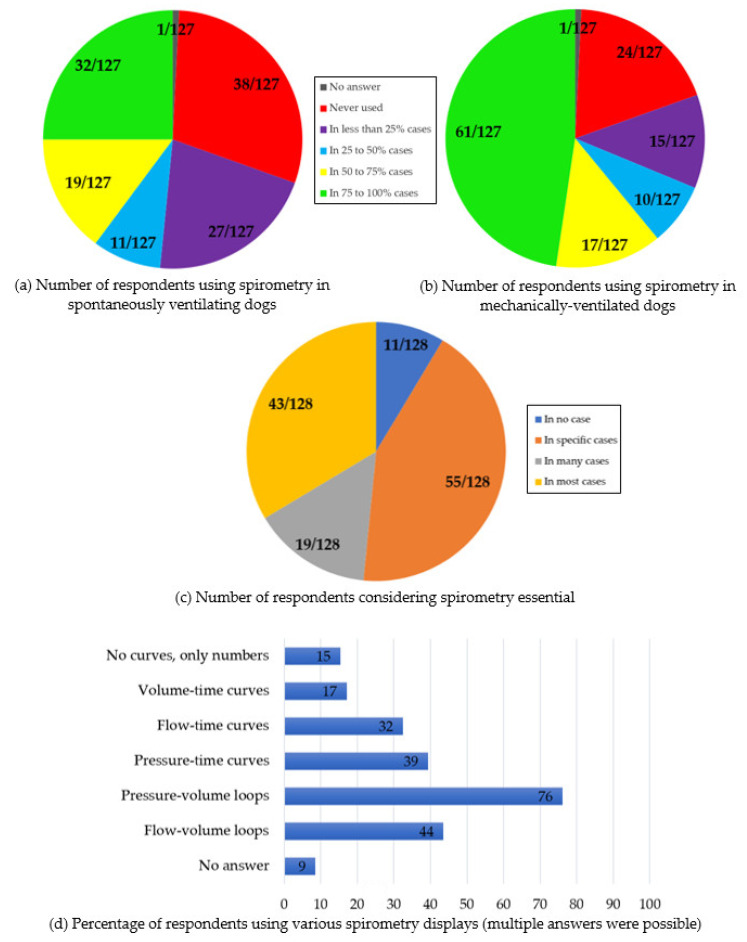
Summary of selected responses to an online survey on the use of spirometry in small animal anaesthesia and critical care by 128 respondents. (**a**) Number of respondents using spirometry in spontaneously ventilating dogs; (**b**) Number of respondents using spirometry in mechanically ventilated dogs; (**c**) Number of respondents considering spirometry essential; (**d**) Percentage of respondents using various spirometry displays.

**Table 1 animals-12-00239-t001:** Demographics of the 128 respondents to a survey on the use of spirometry in small animal anaesthesia and critical care.

Background			
Anaesthetists		Intensivists	
Number		Number	
10 57	Technicians/anaesthesia nurses ACVAA/ECVAA Diplomates	15 1	ACVECC/ECVECC Diplomates ACVECC and ABVP Diplomate, and
			FCCM
2	Post ACVAA/ECVAA residency *(_not Diplomates_)*		
15	ACVAA/ECVAA residents	1	ACVECC/ECVECC resident
13	GP with interest in anaesthesia	2	GP with interest in ECC
1	PhD student in anaesthesia		
Number			
1	ACVECC/ECVECC Diplomate, ACVAA/ECVAA resident		
1	ACVECC/ECVECC and ACVAA/ECVAA resident		
4	GP with interest in both anaesthesia and ECC		
5	Respondents being veterinarians with university teaching roles in anaesthesia and/or ECC but who did not match the abovementioned categories (one had a PhD)		
Country			
United Kingdom (34/128) United States of America (30/128) Australia (13/128) Spain (12/128) Switzerland (10/128) Italy (9/128) Austria (4/128) France (4/128) Canada (3/128) Argentina (2/128) Colombia (1/128) Germany (1/128) New-Zealand (1/128) Norway (1/128) South Africa (1/128) South Korea (1/128) Sweden (1/128)			
Working environment			
University Teaching Hospitals (69/128, 54%) Referral centres (37/128, 29%) Private practice (17/128, 13%) Research institutions (4/128) Telemedicine (1/128).			

ABVP: American Board of Veterinary Practitioners; ACVAA: American College of Veterinary Anaesthesia and Analgesia; ACVECC: American College of Veterinary Emergency and Critical Care; ECC: Emergency and Critical care; ECVAA: European College of Veterinary Anaesthesia and Analgesia; ECVAA: European College of Veterinary Anaesthesia and Analgesia; FCCM: Fellow in the American College of Critical Care Medicine (ACCM); GP: General Practitioner.

## Data Availability

The data is available on reasonable request to the corresponding author.

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
