# Peer review of "A Survey on the Use of Spirometry in Small Animal Anaesthesia and Critical Care"

_animals, 2022, doi:10.3390/ani12030239_

Round 1
Reviewer 1 Report
£45: is this the only and most appropriate reference available?
£48: what does the "this" refer to?
£51: very long sentence difficult to understand
£55: what does "most" mean this is not very scientific
£56: this sentence does not add anything to the study, moreover it underlines the selection bias of the participants
£59: It is good to use recommendations but it would be important to explain their origin (an individual publication in a journal and not from a focus group nor validated to my knowledge)
£60: clarify what is meant by "convenience sample
£61: perhaps point out here that the following chapters respond to the checklist of these recommendations
£63: what does this mean? Study number if submitted? and if not why?
£77: How?
£88: Is all this necessary?
Can you describe the study as open or closed?
£97-101: this does not follow the recommendations. Can you discuss the consequences?
£117 Consequences?
£155: how many did not use spirometry at all (whether the dog was spontaneously breathing or ventilated)? Shouldn't the denominator only count people using it?
£158: a total of 90 users or did some use both?
£160: 23 more so 113 respondents /128?
Generally speaking, a diagram supporting these results would be very useful to better understand the figures
£193: this should be placed at the beginning of the paragraph
£227: more than ?
£219: what is the point of this sentence?
£218-232: this paragraph is not obvious to understand, maybe rewrite it
£247: Yes!!
£249: agreed but it is not obvious from the results
£258: depending on the number of dogs ventilated or left on spontaneous ventilation, this conclusion may be wrong
£260: can be replaced by "therefore it might be interesting to...".
Author Response
- £45: is this the only and most appropriate reference available?
No other relevant reference was found to illustrate this aspect of the introduction. Would the Reviewer have a better suggestion?
- £51: very long sentence difficult to understand:
The sentence was rephrased.
- £55: what does "most" mean this is not very scientific:
The sentence was rephrased.
- £59: It is good to use recommendations but it would be important to explain their origin (an individual publication in a journal and not from a focus group nor validated to my knowledge):
We agree with the reviewer, but this statement is the one that should be followed, in absence of better guidelines, at least in the journal of Veterinary Anaesthesia and Analgesia, one of the references of our community. See a 2020 Editorial from Rachel Bennett, EIC: https://www.vaajournal.org/article/S1467-2987(19)30331-9/fulltext
The statement is logical and represents a decent framework to build a study on.
- £60: clarify what is meant by "convenience sample: ”
Convenience sampling is defined as a method adopted by researchers where they collect market research data from a conveniently available pool of respondents. It is the most commonly used sampling technique as it’s incredibly prompt, uncomplicated, and economical. In many cases, members are readily approachable to be a part of the sample.”
Would the Reviewer recommend to add this information to the manuscript?
- £61: perhaps point out here that the following chapters respond to the checklist of these recommendations:
We leave this decision to the Editors as we feel if is redundant with the previous sentence stating that the CHERRIES statement was followed.
- £63: what does this mean? Study number if submitted? and if not why?
The following sentence was added:
The study was conducted according to the guidelines of the Declaration of Helsinki. Ethical approval was waived for this study after review from the local ethical committee (Gesundheits-, Sozial- un Integrationsdirektion Kantonale Ethikkommission für die Forschung, Kanton Bern) because this project was not subject to any authorization according to national law (Humanforschungsgesetz, Art. 2, Abs 1).
- £77: How?
This comment refers to an empty line in the original manuscript; we don’t undertand what the question refers to and cannot adjust this comment
- £88: Is all this necessary?
We believe it is: it is gives a fair description of the population of respondents we attempted to reach.
- Can you describe the study as open or closed?
Apologies, we don’t understand the question. Does the Reviewer refer to “open-ended” vs “closed-ended” questions? If so, the information was already provided “There were eight sections (one screen each), composed of two to eight items (multiple choice or open questions, all in English). Items were not randomized. Adaptive questioning was not used although some questions were facultative.”
- £97-101: this does not follow the recommendations. Can you discuss the consequences?
Apologies, we don’t understand how this section doesn’t follow “the recommendations” nor which recommendations the Reviewer is referring to, particularly after raising concerns on the CHERRIES statement. In terms of reporting, the information listed here matches the CHERRIES checklist. We find “discussing the consequences” inappropriate in the M&M section, have reported several limitations in the discussion and are happy to add other limitations.
- £117 Consequences?
Apologies, we don’t understand this comment.
- £155: how many did not use spirometry at all (whether the dog was spontaneously breathing or ventilated)? Shouldn't the denominator only count people using it?
This is a very valid point. 24 respondents answered that they never used spirometry but did comment on curves/loops used. This is in fact mentioned in our limitations. We have left the denomitator as “number of respondents” since most people did answer the questions although they claimed they didn’t use the technology.
The information of the total number of people never using spirometry was added in the results.
- Generally speaking, a diagram supporting these results would be very useful to better understand the figures
A figure summarizing key results has been added as requested.
- £158: a total of 90 users or did some use both?
- £160: 23 more so 113 respondents /128?
There seems to be a line issue, the comments are not matching the right sentences of the document. We used the original document to check but this is not the right solution either. Using the search tool in the manuscripts, we didn’t find 90, 23 or 113 in the numbers we reported so cannot adjust those comments. Could the Reviewer be more specific in the questions?
- £227: more than ?
“than static curves” was added.
- £218-232: this paragraph is not obvious to understand, maybe rewrite it
The paragraph has been reworded as requested.
- £249: agreed but it is not obvious from the results
We agree with the reviewers. This is part of the discussion and a reference is added to this statement.
- £258: depending on the number of dogs ventilated or left on spontaneous ventilation, this conclusion may be wrong
We agree with the Reviewers. This is a discussion point.
- £260: can be replaced by "therefore it might be interesting to...".
The sentence has been reworded as suggested.
Reviewer 2 Report
I think this paper is an excellent starting point for a series of studies on the use and misuse of spirometry in veterinary anaesthesia and intensive care.

Author Response
We would like to thank the Reviewer for a positive and constructive review. The manuscript was adjusted accordingly. We hope those changes will match the Reviewer’s expectation.
- The summary and abstract are both good representatives of the paper overall. The only suggestion I would make is to be more explicit at the outset what the ‘community’ is that the authors are referring to e.g line 14.
The sentence was modified as follows: “The way the Veterinary Anaesthesia and Intensive Care communities use spirometry was not found in the literature.”
- I thought the materials and methods were on the whole very clear and would allow the study to be repeated. I was slightly confused in section 2.6 line 97. Was this intended to be a subtitle or separate section? As it stands the sentence does not make sense.
Thank you for this comment. The start of the paragraph has been reworded; we hope the Reviewer will find this version clearer.
- The lay out of Table 1 needs attention as the numbers and groups of respondents do not line up. This may be a reflection of formatting rather than the fault of the authors but it does need attention.
Thank you very much for pointing this out: the automatic formating of the template did alter the display of the table and we didn’t see it. The format has now been altered and the table should be clear. Thank you again!
- I would also be more explicit in the results from each questionnaire section by using sub titles to report the results from each section. Additionally tables would make the results clearer rather than the plain text.
A figure summarizing selected key results was added. We attempted to add subtitles but they weren’t matching the structure of the M&M section and each contained a very short text which, in our opinion, didn’t facilitate the reading. We don’t feel strongly on this point and are happy to amend the text if the Reviewer finds it justified.
- I thought this was well reasoned and reflective on the results, whilst citing the appropriate literature. The overwhelming impression I am left with from this paper is that spirometry is a monitoring aid which is underused and not fully understood by those veterinarians using the technique.
Thank you very much!
Reviewer 3 Report
Dear Authors,
Thank you for submitting your manuscript. It is interesting, well written and well presented. My only suggestion is that it would be beneficial for the readers to include some graphs, i.e. bar graphs, pie charts, etc. in order your figures to be more comprehensive.
Author Response
We would like to thank the Reviewer for a positive and constructive review. The manuscript was adjusted accordingly. We hope those changes will match the Reviewer’s expectation.
My only suggestion is that it would be beneficial for the readers to include some graphs, i.e. bar graphs, pie charts, etc. in order your figures to be more comprehensive.
As requested, a figure summarizing selected key results was added.
Reviewer 4 Report
Dear Authors,
Thanks for having submitted this well designed and interesting study to the present journal. The manuscript is very well written, easy to follow, results clearly stated and supported by data.
I have only few suggestions that may improve the clarity of the manuscript.
1) I think you should report, probably as appendix, the information for participants and the informed consent form. I would also quote them e.g., appendix 1,2...in the text.
2) I think you should also report the original survey again as appendix, this would help the readers to understand better the results.
3) I found the description of the survey a bit vague. Please report at the beginning of section 2.7 how may questions were in total, how many questions where mandatory, how many open-ended (to me sounds better than 'open question'), how many multiple choice and please report which type, I assume single-answer questions, but you should report it).
4) Please also specify if all the multiple choice questions had a DK/No answer options. If not please report as limitations in your discussions.
5) Particularly, please specify how many demographic questions were asked and if there was an other/ DK/No option.
In regard of the discussion, I was wondering if you should mention or not if a similar study has been done in human medicine and what were the results, and/or if this is a first survey in your best knowledge about this topic.
Have a lovely Christmas
Author Response
We would like to thank the Reviewer for a positive and very constructive review. The manuscript was adjusted accordingly. We hope those changes will match the Reviewer’s expectation.
We wish you a very happy festive season!
- I think you should report, probably as appendix, the information for participants and the informed consent form. I would also quote them e.g., appendix 1,2...in the text.
I think you should also report the original survey again as appendix, this would help the readers to understand better the results.
The sentence “The questionnaire is available on request to the corresponding author.” was added instead of following the Reviewer’s suggestion of adding consent form and survey as appendices. Some of the information asked in the survey was used to develop a data collection form to establish reference intervals of compliance and resistance but isn’t reported in the present publication as it doesn’t fall within its scope. We are very happy to provide the questionnaire and rational for not including certain responses in the present manuscript on request, but don’t want to add extra words and confusion here. We hope the Reviewer will find this acceptable.
- I found the description of the survey a bit vague. Please report at the beginning of section 2.7 how may questions were in total, how many questions where mandatory, how many open-ended (to me sounds better than 'open question'), how many multiple choice and please report which type, I assume single-answer questions, but you should report it).
The description has been slightly expanded as requested. However, number of questions and the rest of the Reviewer’s request were not addressed as, as mentioned in the previous comment, many questions were not directly pertinent to this publication and were used for another purpose. This includes in particular type of equipment used, technical details of the equipment, and some variables perceived by the respondents to influence compliance and resistance of the respiratory system. The information was used to design a multicentre data collection, develop the right calibration check protocols and create a suitable data collection form. The data was meaningful but irrelevant here.
- Please also specify if all the multiple choice questions had a DK/No answer options. If not please report as limitations in your discussions.
Questions had a DK/No answer where appropriate and/or weren’t mandatory.
- Particularly, please specify how many demographic questions were asked and if there was an other/ DK/No option.
This information was added in the text.
- In regard of the discussion, I was wondering if you should mention or not if a similar study has been done in human medicine and what were the results, and/or if this is a first survey in your best knowledge about this topic.
Thank you for this comment. We haven’t found any equivalent study in human medicine.
Round 2
Reviewer 1 Report
Dear Authors
thank you for your answers